# Hepatitis E Virus in Croatia in the “One-Health” Context

**DOI:** 10.3390/pathogens10060699

**Published:** 2021-06-04

**Authors:** Anna Mrzljak, Lorena Jemersic, Vladimir Savic, Ivan Balen, Maja Ilic, Zeljka Jurekovic, Jadranka Pavicic-Saric, Danko Mikulic, Tatjana Vilibic-Cavlek

**Affiliations:** 1Department of Gastroenterology and Hepatology, University Hospital Center Zagreb, 10000 Zagreb, Croatia; 2School of Medicine, University of Zagreb, 10000 Zagreb, Croatia; tatjana.vilibic-cavlek@hzjz.hr; 3Department of Virology, Croatian Veterinary Institute, 10000 Zagreb, Croatia; jemersic@veinst.hr; 4Poultry Center, Croatian Veterinary Institute, 10000 Zagreb, Croatia; v_savic@veinst.hr; 5Department of Gastroenterology and Endocrinology, General Hospital “Dr. Josip Bencevic”, 35000 Slavonski Brod, Croatia; ibalen@bolnicasb.hr; 6Department of Epidemiology, Croatian Institute of Public Health, 10000 Zagreb, Croatia; maja.ilic@hzjz.hr; 7Department of Medicine, Merkur University Hospital, 10000 Zagreb, Croatia; zeljka.jurekovic@gmail.com; 8Department of Anestesiology, Reanimatology and Intensive Care, Merkur University Hospital, 10000 Zagreb, Croatia; jpavicic58@gmail.com; 9Department of Abdominal and Transplant Surgery, Merkur University Hospital, 10000 Zagreb, Croatia; danko.mikulic@zg.t-com.hr; 10Department of Virology, Croatian Institute of Public Health, 10000 Zagreb, Croatia

**Keywords:** hepatitis E virus, One-Health, Croatia, epidemiology, human, animal, serology, molecular testing, phylogenetic analysis

## Abstract

Hepatitis E virus (HEV) is the most common cause of viral hepatitis globally. The first human case of autochthonous HEV infection in Croatia was reported in 2012, with the undefined zoonotic transmission of HEV genotype 3. This narrative review comprehensively addresses the current knowledge on the HEV epidemiology in humans and animals in Croatia. Published studies showed the presence of HEV antibodies in different population groups, such as chronic patients, healthcare professionals, voluntary blood donors and professionally exposed and pregnant women. The highest seroprevalence in humans was found in patients on hemodialysis in a study conducted in 2018 (27.9%). Apart from humans, different studies have confirmed the infection in pigs, wild boars and a mouse, indicating the interspecies transmission of HEV due to direct or indirect contact or as a foodborne infection. Continued periodical surveys in humans and animals are needed to identify the possible changes in the epidemiology of HEV infections.

## 1. Introduction

Hepatitis E virus (HEV), a nonenveloped RNA virus, is the most common cause of viral hepatitis with an estimated 939 million of people who have experienced HEV infection [1]. So far, eight genotypes are known: genotypes 1 and 2 are indigenous to humans, while 3–8 are zoonotic genotypes [2]. In developing countries, genotypes 1 and 2 are primarily causing waterborne epidemics. In industrialized countries in Europe, predominantly genotype 3 and, less frequently, genotype 4 are causing zoonotic and foodborne infections in humans [3]. Those genotypes are isolated from a number of animal species, domestic and wildlife (pigs, wild boars, sheep, deer, mongoose, rabbits, goat, yak and horse) [4,5]. The zoonotic potential of HEV genotypes 3 and 4 has been well-demonstrated for pigs [6] and wild boars [7], as well as for genotype 3 in rabbits [8], deer [9] and goats [10] and genotype 4 in cows [11]. Genotypes 5 and 6 are found only in wild boars [12], while genotypes 7 [13] and 8 [14] are derived from camels. Some HEV strains are species-specific, but other strains could cross species and infect many hosts. The cell culture system has been established for the genotype 3 and 4 HEV strains, but genotype 1 replicate poorly in vitro [15]. However, HEV genotype 1 can replicate efficiently in primary cells [16,17,18,19,20] or in vivo animal models [21]. Since pigs and nonhuman primates show similar postinfection dynamics as infected humans; they are suitable models for investigating acute and chronic HEV infections and the propagation, replication and interspecies transmission of the virus [22,23]. In addition, new small animal models have been developed to facilitate the progress in HEV research. However, each model has its applications, advantages and limitations [15].

The consumption of raw and undercooked meat (mainly pork products) [24,25] parenteral blood-based products [26] and solid organ transplants [27] are defined transmission routes in humans. In addition, it is well-established that occupational exposure to known HEV animal reservoirs plays an additional role in the HEV transmission [28]. The results of genotyping analyses show a close relation among all derived isolates, confirming interspecies transmission or a mutual source of infection [29,30]. In humans living in industrialized countries, HEV infection usually presents as an acute mild self-limiting illness; nevertheless, patients with pre-existing [31], as well as patients without pre-existing, liver disease [32] can develop liver failure. In immunocompromised patients, HEV genotype 3 can cause chronic HEV infection with the progression to liver fibrosis [33,34]. In addition, extrahepatic manifestations of HEV have been increasingly recognized with neurological, hematological, gastrointestinal and renal manifestations being the most common [35,36]. In pregnant women with HEV infection, fulminant hepatic failure (FHF) may develope, especially in the second and third trimesters, with high maternal mortality [37,38,39]. Serology and nucleic acid testing in combination are used to confirm the diagnosis of HEV infection. However, the serological results must be interpreted with caution due to the diversity of serology tests used with different diagnostic sensitivity and specificity.

In a zoonotic infection such as HEV, it is of the utmost importance to recognize the interconnections between humans, animals, and their shared environment and to implement a “One-Health” multisectoral approach to achieve optimal health and outcomes. The zoonotic HEV genotype 3 in Southeast Europe [40] HEV has been increasingly detected in animals (pigs [40,41,42,43] and wild boars [44,45]) and human populations [46,47,48,49], supporting the need for a closer look in the region. This narrative review comprehensively addresses the current data on the HEV epidemiology in humans and animals in Croatia.

## 2. Methods

We searched PubMed, Web of Science, Medline, Scopus, ScienceDirect and ResearchGate with no limitations of the year of publication nor language restriction using a predefined search strategy “hepatitis E virus”, “human”, “animal”, “environment” and “Croatia”. Once a comprehensive list of abstracts was retrieved and reviewed, studies appearing to meet inclusion criteria were reviewed in full. Studies nonrelevant for the topic or the ones with data inconsistency were also excluded, as assessed by the authors. Books, dissertations and unpublished reports were excluded.

## 3. Hepatitis E Virus in Croatia–Human Studies

The first case of autochthonous HEV infection in Croatia was reported in 2012 with the undefined zoonotic transmission of HEV genotype 3 [50]. Consequently, the first HEV prevalence study conducted from 2011 to 2013 among 504 hepatitis patients negative for acute viral hepatitis A-C, and 88 HIV-infected patients showed an anti-HEV IgG seroprevalence of 8.7% and 1.1%, respectively (Figure 1). Anti-HEV IgM prevalence among hepatitis and HIV patients was 3.2% and 1.1%, respectively. HEV RNA was detected in five of fourteen anti-HEV IgM-positive patients, and a phylogenetic analysis confirmed genotype 3. None of the tested patients had a recent history of travel to disease endemic areas, and infections were presumed autochthonous [51].

Later on, a study analyzing the HEV seroprevalence in different population groups was carried out from 2014 to 2015. The overall HEV IgG and IgM seropositivity in the 214 tested was 5.6% and 1.9%, respectively, with no HEV RNA detected. Alcohol abusers (8.9%) and war-related post-traumatic stress disorder patients (8.6%) had the highest HEV IgG seroprevalence, followed by persons who inject drugs (6.1%) and healthcare professionals (2.7%). The study demonstrated that HEV IgG seropositivity increased with age (2.3% for <40 years vs. 11.3% for >50 years), number of household members (1.8% for 2 members vs. 12.1% for >4 members) and rural area of residence (14.5% for rural and suburban vs. 2.5% for urban areas) [52].

In addition, there are several published articles on the HEV seroprevalence in Croatia’s general population groups. As in many other European countries, HEV testing of blood donors in Croatia is not mandatory. However, in 2014, serum samples from 1036 voluntary blood donors from six counties showed surprisingly high HEV IgG seropositivity (20.2%), although HEV RNA was not detected in any of 4.4% HEV IgM-reactive samples. As confirmed previously, a lower seroprevalence was found in blood donors younger than 40 years compared to the older ones. In addition, a lower HEV IgG prevalence was observed in urbanized Zagreb County (7.5%) compared to rural Bjelovar-Bilogora County (50.3%) [53]. Only one study addressed the HEV seroprevalence in pregnant women in Croatia, showing previous exposure in 2.9% of women [54].

Data on professional exposures in Croatia are scarce and conducted on small cohorts. In 2016 and 2017, forest workers (8.1%) and hunters (4.0%) showed higher HEV IgG seropositivity compared with the general population (3.4%) [55].

In the context of chronic diseases, strong evidence supports that HEV infection is more prevalent in patients on hemodialysis compared with non-hemodialysis control groups [56]. In 2018, a seroprevalence study [57] conducted among 394 hemodialysis patients from continental and coastal Croatian regions showed an overall anti-HEV IgG prevalence of 27.9%, suggesting a high degree of previous exposure to HEV. There were significant variations between regions, with the highest (43.4%) being in the continental county around the city of Osijek, the region densely populated with pig farms. On the other side of the spectrum, hemodialysis patients in Dubrovnik-Neretva County, coastal and the most southern region, had a seroprevalence of only 5.2%. The authors speculated that several environmental and cultural factors were responsible for the regional differences, particularly traditional pork-based food in the continental part. The study confirmed that the seroprevalence increased with age, from 5.3% in patients less than 40 years old to 31.2% in patients older than 60 years. However, no significant difference was found in HEV IgG seropositivity according to gender, level of education, hemodialysis duration, a rural or urban area of residence, eating habits, profession, traveling habits and source of drinking water or the type of sewage system. The study identified several independent factors for IgG seropositivity: age > 60 years (OR = 8.17; 95% CI = 1.08–62.14), living in the continental region (OR = 2.58; 95% CI = 1.55–4.30) and being a recipient of blood products transfusion (OR = 1.66; 95% CI = 1.01–2.73). HEV IgM seropositivity was found in 3.7% of IgG-positive hemodialysis patients, and HEV RNA was not detected [57].

A recently published study conducted on a large number of adult patients (*n* = 438) with various chronic liver diseases (2016–2018) demonstrated that the HEV burden in Croatia is high. The previous exposure to the HEV was detected in 15.1% of patients, with 4.5% being IgM-positive, but not a single one tested HEV RNA-positive. The study again substantiated that the seroprevalence increases with age, from 9.7% in younger than 45 years to 17.4% in older than 60 years. No differences in HEV-IgG seropositivity related to gender, level of education, geographic region, area of residence, liver disease or hepatocellular carcinoma presence were detected [58].

To date, only one study addressed the HEV seroprevalence in the Croatian transplant recipients [59]. In 2017, a cross-sectional study conducted in a cohort of 242 adult liver transplant recipients, transplanted from 1994 to 2013 in a single high-volume liver transplant center, showed an HEV IgG seroprevalence rate of 24.4%. HEV IgM antibodies were found in only one recipient, and one patient showed an equivocal result, but HEV RNA was not detected. The study identified several independent risk factors for HEV seropositivity, such as female gender, older age and sewage system connected to a septic tank, whereas the highest level of education was identified as a protective factor.

## 4. Hepatitis E Virus in Croatia—Animal Studies

In Croatia, the first samples of animal origin were tested for the presence of HEV RNA in 2007; however, a prospective large-scale animal testing study started in 2009 [60]. The results of genotyping analyses showed that subtype 3a is predominant in Croatia (65.5%) and is derived from humans, pigs, wild boars and a mouse. Subtype 3c is also highly represented (23.1%) and isolated from humans and pigs, while subtypes 3e and 3f are found sporadically: 3e in humans and pigs, whereas 3f only in humans (Figure 2). According to the results, subtypes 3a and 3c can be considered endemic in Croatia, while subtypes 3e and 3f indicate the possibility of newly imported infections. Since homologous strains have been found in humans, pigs, wild boars and a mouse, the interspecies transmission of HEV due to direct or indirect contact or as a foodborne infection cannot be excluded [60].

### 4.1. Pigs

Pigs are natural hosts of HEV and are considered to be the most important animal reservoirs of the disease in humans. The infection in pigs is subclinical, with rare occurrences of microscopic histopathological lesions of the liver [61].

The Croatian domestic pig population encounters are 1,398,556 (Croatian Agriculture Agency, Ministry of Agriculture). However, the distribution of pig farms differs among the regions where 98% of all farms are located in continental Northern Croatia. High pig density; type of farm management (large farms with appropriate bio-security measures implemented, ecological management of breeding, or small-scale breeding farms) and a possibility of contact with the wildlife, primarily wild boars, are considered as risk factors for HEV spread to and from pigs [30].

During 2009/2010, the samples of pig origin (blood, spleen and lymph nodes) were tested for the presence of HEV RNA. The overall RNA prevalence of 24.5% was found [29] (Table 1). Most positive findings were detected in small husbandries (‘backyard’ pigs), where 30.5% of the tested pigs were shown to be HEV RNA-positive. A statistically significant higher HEV RNA prevalence was observed in fattening pigs when compared to the other age categories [29]. This result is within expectations due to the duration of viremia in pigs; however, it opens up a concern of HEV entering the food chain. Apart from blood and tissue samples, positive fecal samples were detected regardless of the breeding system, confirming viral shedding in detectable doses via feces and allowing possible environmental contamination, especially in breeding systems with ecological management where pig feces may be used as manure [63]. Further studies [30] carried out until 2017 showed a decrease in HEV RNA prevalence in pigs (15.2%) when compared to previously conducted studies [29]. Since most of the pigs included in this study were older than four months of age, a lower prevalence of viremia was expected; however, this result may be a consequence of a recycling HEV infection within the investigated farms. A study regarding the seroprevalence of HEV in pigs was carried out and a seropositivity of 32.94% was determined [63]. Cumulative HEV RNA prevalence in the counties of Croatia is presented in Figure 3a.

The introduction of HEV infection that probably occurred in pigs in Croatia from 2007 onwards has not yet been entirely revealed; however, due to the results of genetic analyses from 2009 to 2012 and detecting genotypes that show high genetic analogies with isolates from other European countries, it is presumed that the virus was introduced by international trade. Furthermore, homologous strains are found in humans, wild boars and a mouse, indicating that pigs in Croatia present the primary animal reservoir of HEV.

### 4.2. Wild Boars

Since big game hunting is an important activity in Croatia, direct or indirect contact among hunters and wild boars is inevitable. According to the hunting bag, the estimated number of wild boars in Croatia varies from 45,700 to 65,000 per year. The highest density of the wild boar population (98%) is within the central and northern regions. However, as a result of the lack of natural enemies, as well as preferable climate changes and forest management, the habitat of wild boars is spreading due to their increasing number. Currently, wild boars have spread from the north of Croatia to the coastal regions, even inhabiting islands [51]. The high density of wild boars, direct or indirect contact with HEV-positive domestic pigs, and a contaminated environment are recognized as potential risk factors for the spread of HEV among the wild boar population [29,63,64]. Infected wild boars shed the virus via feces, ensuring further viral spread [65].

From 2009, HEV RNA is consecutively derived from wild boars showing a consistent viral prevalence of 12.30% [29], 11.33% [63], and 11.50% [30]. The cumulative HEV RNA prevalence in the counties of Croatia is presented in Figure 3b. The overall seroprevalence found was 31.10%. Interestingly, seropositive wild boars younger than one year of age showed to be simultaneously HEV RNA-positive, indicating chronic infection and a possible prolonged virus spread. The detected viral presence despite the immune response confirms that wild boars have a key role in HEV environmental maintenance [63] and may directly be responsible for virus spread to other wildlife species.

### 4.3. Birds

Avian HEV strains belong to the *Orthohepevirus B* genera and causes infection with a subclinical course or a clinically manifested disease referred to as hepatitis–splenomegaly syndrome in chickens [66,67]. One Croatian study revealed that chicken flocks tested HEV RNA-negative [68]. Further epidemiological investigations are needed to determine the importance of birds in HEV spread and transmission.

### 4.4. Shellfish

Shellfish can be important indicators of HEV infection, since they accumulate viruses during filtration. In Croatia, 252 mussels (*Mytilus galloprovincialis*) and 286 oysters (*Ostrea edulis*) were tested for a HEV RNA presence during 2009 and 2010. No traces of HEV genome were detected [29].

### 4.5. Other Investigated Species

During the period from 2009 to 2012, a number of animal species in Croatia were investigated for the presence of HEV RNA, such as 32 samples of cattle (*Bos taurus*), 40 samples of roe deer (*Capreolus capreolus*), 280 samples of red deer (*Cervus elaphus*), 12 mouflon samples (*Ovis musimon*), 10 samples of martens (*Martes martes*), 8 skunk samples (*Mustela putorius* Linnaeus) and 50 red fox samples (*Vulpes vulpes*); however, none of the tested samples was found positive [29]. Currently, only one report of a mouse (yellow-necked mouse) naturally infected by genotype 3a is available [69].

## 5. Conclusions and Future Perspectives

This narrative review comprehensively addresses the current knowledge on the HEV epidemiology in humans and animals in Croatia (“One Health” context). There have been several studies assessing the seroprevalence in different population groups. The results indicate that HEV infection is widespread in Croatia: the seroprevalence in voluntary blood donors was 21.5%, significantly higher among people over 40 years old and those living in rural areas. The highest seroprevalence in humans was found in patients on hemodialysis in a study conducted in 2018 (27.9%), with the highest seroprevalence of more than 43% in the continental region around the city of Osijek, the area densely populated with pig farms. Apart from humans, different studies have confirmed the infection in pigs, wild boars and, in one sample, from a mouse. The results of the genotyping analyses showed that subtype 3a is predominant in Croatia (65.5%). Continuous periodical seroepidemiological surveys in different population groups in Croatia are needed to identify possible changes in the epidemiology of HEV infection. In addition, further epidemiological studies are needed to determine the importance of different animals in HEV spread and transmission. Furthermore, more extensive studies are needed to understand the transmission modes between species to formulate public health policies in order to prevent HEV infection in humans.

## Figures and Tables

**Figure 1 pathogens-10-00699-f001:**
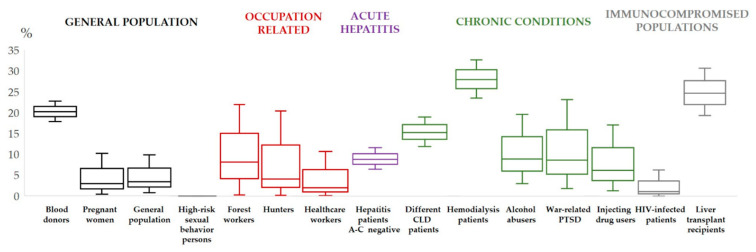
Hepatitis E virus IgG prevalence in different population groups in Croatia [51,52,53,54,55,56,57,58,59].

**Figure 2 pathogens-10-00699-f002:**
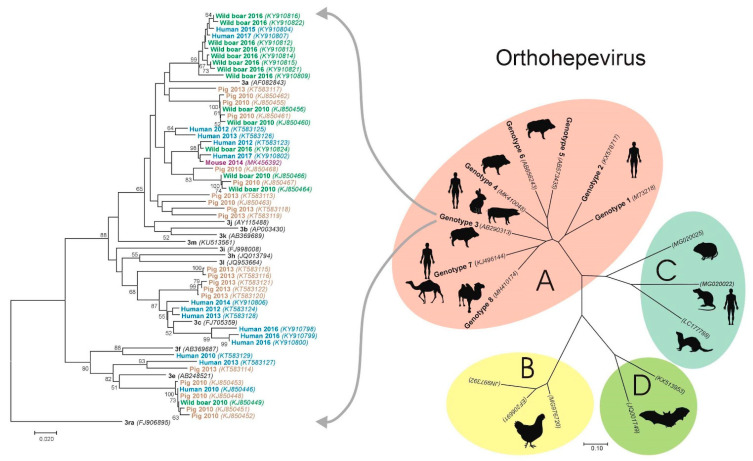
The radial phylogenetic tree displays genus *Orthohepevirus* comprising four species: *Orthohepevirus A*, *B*, *C* and *D*, respectively, with the main hosts indicated as figures. The rectangular phylogenetic tree displays the genetic and host diversity of the selected hepatitis E virus (HEV) isolates from Croatia, which belong to genotype 3 of *Orthohepevirus A*, as indicated in the radial phylogenetic tree. The evolutionary history was inferred using the Neighbor-Joining method. Supporting (≥50%) bootstrap values of 1000 replicates are displayed at the nodes in the rectangular tree. Scale bars indicate nucleotide substitutions per site. HEV isolates from Croatia are color-coded (blue—human origin, green—wild boar origin, brown—pig origin and purple—mouse origin). Designations also include detection years for Croatian strain and GenBank accession numbers. The taxa in black color in the rectangular tree are genotype 3 subtype reference strains, according to Smith et al. [62].

**Figure 3 pathogens-10-00699-f003:**
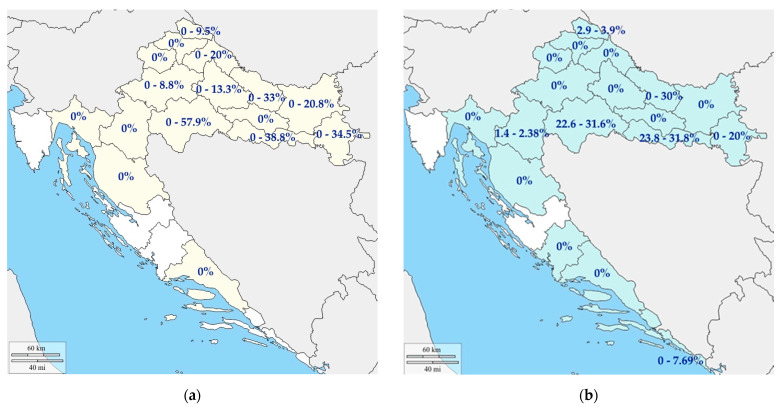
HEV RNA prevalence in Croatia for: (**a**) domestic pigs and (**b**) wild boars.

**Table 1 pathogens-10-00699-t001:** Epidemiology of hepatitis E in domestic pigs and wild boars in Croatia.

Species	Years of Investigation	*N* Tested	HEV RNA Prevalence % Positive (95% CI)	HEV IgG Prevalence % Positive (95% CI)	Reference
Domestic pigs	2010–2017	1419	15.2% (13.5–17.2)	ND ^1^	[30]
	2016 and 2017	1424	NT ^2^	32.9% (30.5–35.4%)	[63]
	2016 and 2017	670	0	NT ^2^	[63]
	2009 and 2010	1092	24.5% (21.7–27.6%)	ND ^1^	[29]
Wild boars	2010–2017	720	11.5% (9.4–14.1%)	ND ^1^	[30]
	2016 and 2017	1000	-	31.1% (28.3–34.0%)	[63]
	2016 and 2017	150	31.10% (28.31–34.04%)	NT ^2^	[63]
	2009 and 2010	536	12.3% (9.7–15.4%)	ND ^1^	[29]

^1^ ND—not detected; ^2^ NT—not tested.

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
