# Peer review of "Hepatitis E Virus in Croatia in the “One-Health” Context"

_pathogens, 2021, doi:10.3390/pathogens10060699_

Round 1

Reviewer 1 Report

Reviewer Comments to Authors

The idea of the manuscript is very good. The manuscript is very well written. I think this manuscript will be interesting to readers of a scientific journal "Pathogens" and for the scientific community. However, I have recommendations which I think will increase the quality of this manuscript. The recommendations are shown below.   

(1.) Introduction: I think that is very good in this section to add information (3-4 sentences) for HEV infection in other countries in the region (countries of Southeastern Europe). In this regard, I recommend to add the following scientific publications to this section (and add to section "References"):   

  1. Zele D, Barry AF, Hakze-van der Honing RW, Vengust G, van der Poel
    WH. Prevalence of anti-hepatitis E virus antibodies and first detection
    of hepatitis E virus in wild boar in Slovenia. Vector Borne Zoonotic
    Dis. 2016; 16(1): 71-74. [DOI: 10.1089/vbz.2015.1819] [PMID: 26757050]

    2. Tsachev I, Baymakova M, Ciccozzi M, Pepovich R, Kundurzhiev T,
    Marutsov P, Dimitrov KK, Gospodinova K, Pishmisheva M, Pekova L.
    Seroprevalence of hepatitis E virus infection in pigs from Southern
    Bulgaria. Vector Borne Zoonotic Dis. 2019; 19(10): 767-772. [DOI:
    10.1089/vbz.2018.2430] [PMID: 31017536]

    3. Petrovic T, Lupulovic D, Jimenez de Oya N, Vojvodic S, Blazquez AB,
    Escribano-Romero E, Martin-Acebes MA, Potkonjak A, Milosevic V, Lazic S,
    Saiz JC. Prevalence of hepatitis E virus (HEV) antibodies in Serbian
    blood donors. J Infect Dev Ctries. 2014; 8(10): 1322-1327. [DOI:
    10.3855/jidc.4369] [PMID: 25313610]

    4. Tsachev I, Baymakova M, Pepovich R, Palova N, Marutsov P, Gospodinova
    K, Kundurzhiev T, Ciccozzi M. High seroprevalence of hepatitis E virus
    infection among East Balkan swine (Sus scrofa) in Bulgaria: Preliminary
    results. Pathogens. 2020; 9(11): 911. [DOI: 10.3390/pathogens9110911]
    [PMID: 33153218]

    5. Milojevic L, Velebit B, Teodorovic V, Kirbis A, Petrovic T, Karabasil
    N, Dimitrijevic M. Screening and molecular characterization of hepatitis
    E virus in slaughter pigs in Serbia. Food Environ Virol. 2019; 11(4):
    410-419. [DOI: 10.1007/s12560-019-09393-1] [PMID: 31243738] 

    6. Porea D, Anita A, Demange A, Raileanu C, Oslobanu Ludu L, Anita D,
    Savuta G, Pavio N. Molecular detection of hepatitis E virus in wild boar
    population in Eastern Romania. Transbound Emerg Dis. 2018; 65(2):
    527-533. [DOI: 10.1111/tbed.12736] [PMID: 29027370]

    7. Baymakova M, Terzieva K, Popov R, Grancharova E, Kundurzhiev T,
    Pepovich R, Tsachev I. Seroprevalence of hepatitis E virus infection
    among blood donors in Bulgaria. Viruses. 2021; 13(3): 492. [DOI:
    10.3390/v13030492] [PMID: 33809748]

    8. Porea D, Anita A, Vata A, Teodor D, Crivei L, Raileanu C, Gotu V,
    Ratoi I, Cozma A, Anita D, Oslobanu L, Pavio N, Savuta G. Common
    European origin of hepatitis E virus in human population from Eastern
    Romania. Front Public Health. 2020; 8: 578163. [DOI:
    10.3389/fpubh.2020.578163] [PMID: 33392130]

    9. Pittaras T, Valsami S, Mavrouli M, Kapsimali V, Tsakris A, Politou
    M. Seroprevalence of hepatitis E virus in blood donors in Greece. Vox
    Sang. 2014; 106(4): 387. [DOI: 10.1111/vox.12122] [PMID: 24387713]

(2.) Materials & Methods: I recommend to add that you did search and in a database "ScienceDirect".   

Author Response

Re: Manuscript Pathogens-1241651

Dear Editor,

thank you for your letter of May 27th and the possibility of resubmitting our revised manuscript titled "Hepatitis E virus in Croatia in the One-Health Context" for consideration for publication in the Pathogens.

We have carefully considered the reviewer’s comments, and revised the manuscript accordingly. The changes are marked using “Track changes” in the revised version of the manuscript. Below are the answers to specific reviewers’ comments.

Reviewer #1

Introduction: I think that is very good in this section to add information (3-4 sentences) for HEV infection in other countries in the region (countries of Southeastern Europe). In this regard, I recommend to add the following scientific publications to this section (and add to section "References"):   

  1. Zele D, Barry AF, Hakze-van der Honing RW, Vengust G, van der Poel Prevalence of anti-hepatitis E virus antibodies and first detection of hepatitis E virus in wild boar in Slovenia. Vector Borne Zoonotic Dis. 2016; 16(1): 71-74. [DOI: 10.1089/vbz.2015.1819] [PMID: 26757050]
  2. Tsachev I, Baymakova M, Ciccozzi M, Pepovich R, Kundurzhiev T, Marutsov P, Dimitrov KK, Gospodinova K, Pishmisheva M, Pekova L. Seroprevalence of hepatitis E virus infection in pigs from Southern Vector Borne Zoonotic Dis. 2019; 19(10): 767-772. [DOI: 10.1089/vbz.2018.2430] [PMID: 31017536]
  3. Petrovic T, Lupulovic D, Jimenez de Oya N, Vojvodic S, Blazquez AB, Escribano-Romero E, Martin-Acebes MA, Potkonjak A, Milosevic V, Lazic S, Saiz JC. Prevalence of hepatitis E virus (HEV) antibodies in Serbian blood donors. J Infect Dev Ctries. 2014; 8(10): 1322-1327. [DOI: 3855/jidc.4369] [PMID: 25313610]
  4. Tsachev I, Baymakova M, Pepovich R, Palova N, Marutsov P, Gospodinova K, Kundurzhiev T, Ciccozzi M. High seroprevalence of hepatitis E virus infection among East Balkan swine (Sus scrofa) in Bulgaria: Preliminary Pathogens. 2020; 9(11): 911. [DOI: 10.3390/pathogens9110911] [PMID: 33153218]
  5. Milojevic L, Velebit B, Teodorovic V, Kirbis A, Petrovic T, Karabasil N, Dimitrijevic M. Screening and molecular characterization of hepatitis E virus in slaughter pigs in Serbia. Food Environ Virol. 2019; 11(4): 410-419. [DOI: 10.1007/s12560-019-09393-1] [PMID: 31243738] 
  6. Porea D, Anita A, Demange A, Raileanu C, Oslobanu Ludu L, Anita D, Savuta G, Pavio N. Molecular detection of hepatitis E virus in wild boar population in Eastern Romania. Transbound Emerg Dis. 2018; 65(2): 527-533. [DOI: 10.1111/tbed.12736] [PMID: 29027370]
  7. Baymakova M, Terzieva K, Popov R, Grancharova E, Kundurzhiev T, Pepovich R, Tsachev I. Seroprevalence of hepatitis E virus infection among blood donors in Bulgaria. Viruses. 2021; 13(3): 492. [DOI: 3390/v13030492] [PMID: 33809748]
  8. Porea D, Anita A, Vata A, Teodor D, Crivei L, Raileanu C, Gotu V, Ratoi I, Cozma A, Anita D, Oslobanu L, Pavio N, Savuta G. Common European origin of hepatitis E virus in human population from Eastern Front Public Health. 2020; 8: 578163. [DOI: 10.3389/fpubh.2020.578163] [PMID: 33392130]
  9. Pittaras T, Valsami S, Mavrouli M, Kapsimali V, Tsakris A, Politou Seroprevalence of hepatitis E virus in blood donors in Greece. Vox Sang. 2014; 106(4): 387. [DOI: 10.1111/vox.12122] [PMID: 24387713]

We thank the reviewer for suggestions. A short overview of HEV infection in other countries of the region has been made according to proposed references.

  1. Materials & Methods: I recommend to add that you did search and in a database "ScienceDirect". 

We searched the ScienceDirect database and added it to Materials and Methods.

We corrected several typos and additionally revised manuscript for grammar and language.

In conclusion, we thank the reviewer for recognizing the presented mini-review as a good scientific effort, as well as for the useful and constructive comments, which made us think more critically about the presentation of our data. We hope that we have improved the consistency, clarity and interpretation of data in the revised manuscript and that the revised manuscript will meet the reviewers’ requirements and be suitable for publication in the Pathogens.

Thank you again for the privilege of submitting our work to the Pathogens.

Sincerely,

Anna Mrzljak, MD, PhD

University Hospital Center Zagreb

School of Medicine, Zagreb, Croatia

Reviewer 2 Report

Mrzljak et al. presented a review article entitled "Hepatitis E virus in Croatia in the “One-Health” Context ". The review focused on the epidemiology and source of HEV infection in Croatia including human studies and animal studies.

The review could be of a scientif merits, especially for the Croats (Croatians). However, I have some comments on the review in the present form

1- My major comment is most information published in this review had been already published by the same group in 2019 in World Journal of Gatsroenterology. PMID: 31333309 , DOI : 10.3748/wjg.v25.i25.3168.

Even about 70-80% of the data in Table 1 of the current review was already mentioned in the previous review PMID: 31333309 , Table 1.

I understand that the current review focuses mainly on Croatia, while the previous one focused on South-East Europe. But still the information is the same. Also some information is not matched even with same cohort. I checked quickly both tables, and I found some prevelance are not the same even in the same study, for example HEV IgG prevalence in Voluntary blood donors (20.3% in one study and 21.5% in another), and HEV IgG prevalence and HEV IgM prevalence in acute hepatitis nonA_C. Please revise the perecentage of prevalence.

I suggest the authors the following

a) Expand the introduction part: include extrahepatic pathogenesis, cell culture model, in vivo animal models,...etc.

b) Include new titles about futrue perspectives of HEV in Croatia

2) The authors need to do extensive revision on the information and references presented in the review

a) Most of the review is extracted from another review, However, the authors need to cite the original research articles for example ref 4, ref5, 6,7, 8, 

b) Some information and references were outdated: for examples : "Hepatitis E virus (HEV), a non-enveloped RNA virus, is the most common cause of viral hepatitis due to an estimated 20 million infections and 70,000 deaths annually [1,2]"

This is very old information, the WHO documented 14 million HEV infections with 300000 deaths and 5200 stillbirths globally/year (https://apps.who.int/iris/handle/10665/206521).

In addition ref 1,2 should be replaced with  a new metanalysis review articl

https://doi.org/10.1111/liv.14468.

c) Some information is not correct

-page 1 line 45: Pigs [5] are the only known natural animal hosts of genotypes 3 and 4 . This is not correct, since wild boar, deer, rabbits are also natural animal hosts. please correct and cite the adequate ref

-page 1-2 line 46-48: show similar post-infection dynamics as infected humans therefore are suitable models for investigating HEV chronic infection, as well as the propagation, replication and interspecies transmission of the virus . Please also consider that acute HEV infection in recorded in pigs , please mention and cite the adquate ref PMID: 25320303

  • page 2 line 53 : Apart from humans, the infection is confirmed in pigs, wild boars and a mouse. Please delete mouse, there are worthy animals to be mentioned in this place such as rabbits, deer. Please cite the adequate ref.
  • line 57: , patients with pre-existing liver disease can develop liver failure. Recent studies showed also acute liver failure could be developed in patients without preexisting liver disease. Please include this information and cite the adequate references PMID: 34002677, PMID: 33469320 .
  • line 58: HEV genotypes 3 and 4 cause chronic HEV infection with the 
    rapid progression of liver fibrosis.  This is not correct, chronic infection could be progressed to liver fiborsis in only minor cases and not rapide progression. Please correct and add adequate references 
    In general the review needs extensive revision on the scientific knoweldge and the cited references. With including more titles to be distinct from the previous published review done by the same group

Author Response

Re: Manuscript Pathogens-1241651

Dear Editor,

thank you for your letter of May 27th and the possibility of resubmitting our revised manuscript titled "Hepatitis E virus in Croatia in the One-Health Context" for consideration for publication in the Pathogens.

We have carefully considered the reviewer’s comments and revised the manuscript accordingly. The changes are marked using “Track changes” in the revised version of the manuscript. Below are the answers to specific reviewers’ comments.

Reviewer #2

My major comment is most information published in this review had been already published by the same group in 2019 in World Journal of Gatsroenterology. PMID: 31333309 , DOI : 10.3748/wjg.v25.i25.3168. Even about 70-80% of the data in Table 1 of the current review was already mentioned in the previous review PMID: 31333309 , Table 1. I understand that the current review focuses mainly on Croatia, while the previous one focused on South-East Europe. But still the information is the same. Also some information is not matched even with same cohort. I checked quickly both tables, and I found some prevelance are not the same even in the same study, for example HEV IgG prevalence in Voluntary blood donors (20.3% in one study and 21.5% in another), and HEV IgG prevalence and HEV IgM prevalence in acute hepatitis nonA_C. Please revise the perecentage of prevalence.

World Journal of Gastroenterology. PMID: 31333309 , DOI : 10.3748/wjg.v25.i25.3168 discusses data on HEV epidemiology in the whole region of South East Europe, as well as in Croatia as its integral part. This review focuses on Croatian data only, exploring in detail human and animal samples. We thank the reviewer for the suggestions regarding Table 1. We have corrected some misspelled data and rearranged Table 1 into Figure 1 to make it more appealing for the readers.

I suggest the authors the following

  1. a) Expand the introduction part: include extrahepatic pathogenesis, cell culture model, in vivo animal models,...etc.

We thank the reviewer for suggestions, we have expanded the introduction part with new information regarding extrahepatic pathogenesis, cell culture model and in vivo animal models in HEV reserch.

  1. b) Include new titles about futrue perspectives of HEV in Croatia

We added future perspective to the title.

2) The authors need to do extensive revision on the information and references presented in the review

  1. a) Most of the review is extracted from another review, However, the authors need to cite the original research articles for example ref 4, ref5, 6,7, 8, 

We made extensive revision of references citing the original articles in this review. As suggested, the majority of references in the Introduction part has been changed.

  1. b) Some information and references were outdated: for examples : "Hepatitis E virus (HEV), a non-enveloped RNA virus, is the most common cause of viral hepatitis due to an estimated 20 million infections and 70,000 deaths annually [1,2]" This is very old information, the WHO documented 14 million HEV infections with 300000 deaths and 5200 stillbirths globally/year. In addition ref 1,2 should be replaced with  a new metanalysis review article https://doi.org/10.1111/liv.14468.

We replaced an estimated number of worldwide infections with newer data for HEV prevalence.

  1. c) Some information is not correct

page 1 line 45: Pigs [5] are the only known natural animal hosts of genotypes 3 and 4. This is not correct, since wild boar, deer, rabbits are also natural animal hosts. Please correct and cite the adequate ref.

The statement about pigs and wild boar, deer and rabits as natural animal hosts for HEV infection is has been corrected.

page 1-2 line 46-48: show similar post-infection dynamics as infected humans therefore are suitable models for investigating HEV chronic infection, as well as the propagation, replication and interspecies transmission of the virus. Please also consider that acute HEV infection is recorded in pigs, please mention and cite the adequate ref. PMID: 25320303.

As requested, the reference about pig model for acute HEV infection has been added.

page 2 line 53 : Apart from humans, the infection is confirmed in pigs, wild boars and a mouse. Please delete mouse, there are worthy animals to be mentioned in this place such as rabbits, deer. Please cite the adequate reference.

Corrected according to the reviewers suggestion

line 57: , patients with pre-existing liver disease can develop liver failure. Recent studies showed also acute liver failure could be developed in patients without preexisting liver disease. Please include this information and cite the adequate references PMID: 34002677, PMID: 33469320 .

We included this recent discovery by a proper citatition in our review article thanks to the valuable remark by the reviewer.

line 58: HEV genotypes 3 and 4 cause chronic HEV infection with the rapid progression of the liver fibrosis. This is not correct, chronic infection could be progressed to liver fibrosis in only inor cases and not rapide progression. Please correct and add adequate references.

We added an adequate reference for the corrected statement involving HEV infection and liver fibrosis.

In general the review needs extensive revision on the scientific knoweldge and the cited references. With including more titles to be distinct from the previous published review done by the same group.

We corrected several typos and additionally revised manuscript for grammar and language.

In conclusion, we thank the reviewers for recognizing the presented mini-review as a good scientific effort, as well as for the useful and constructive comments, which made us think more critically about the presentation of our data. We hope that we have improved the consistency, clarity and interpretation of data in the revised manuscript and that the revised manuscript will meet the reviewers’ requirements and be suitable for publication in the Pathogens.

Thank you again for the privilege of submitting our work to the Pathogens.

Sincerely,

Anna Mrzljak, MD, PhD

University Hospital Center Zagreb

School of Medicine, Zagreb, Croatia

Round 2

Reviewer 2 Report

I checked the revised manuscript submitted by the authors. Most of the issues have been resolved carefully by the authors. I like the idea that the authors replace the table with a graph figure which gives new appearance of the results. 

I found also the authors have included and/or changed, some, not all of the suggested references. 

I have few minor suggestions

a) Introduction: page 1 lines 43-45 " Zoonotic potential of HEV genotypes 3 and 4 has been well demonstrated for pigs [5] and wild boars [6], as well as for the genotype 3 in rabbits [7] and deer [8].

Please also include the other potential animal reservoirs such as cows (PMID: 27286751), goat (PMID: 33322702, PMID: 32659521) and sheep (PMID: 33322702)

Please include the above information and cite the suggested references.

2- Introduction page 2 line 48-49: The cell culture system has been established for genotype 3 and 4 HEV strains, but genotype 1 replicate poorly in vitro [12].,

Please add that. However, HEV genotype 1 can replicate efficiently in primary cells (  PMID: 31727684, PMID: 32316431, PMID: 32455708, PMID: 30420629, PMID: 32824088) or in vivo animal models (PMID: 26038457).

Please add the information with all suggested references

3- Introduction page 2 lines 64-66:  In addition, extrahepatic manifestations of HEV have been increasingly recognized with neurological (55%), cardiovascular or hematological (35%), and gastrointestinal 66
manifestations (7%) being the most common [26]

a) Please delete cardiovascular, instead include renal . In addition please add complication during pregnancy.

b) Please delete all the percentage, they are not true.

c) Please include the 2 other references that cover all extrahepatic manifestation of HEV (PMID: 27913223 )

Please  correct/ add the information with all suggested references

At the end, I want to thank the authors for their patience and including all the suggested comments. Congratulations for this nice review.  

Author Response

Jun 1st, 2021

Re: Manuscript Pathogens-1241651

Dear Editor,

thank you for your letter and the possibility of resubmitting our revised manuscript titled "Hepatitis E virus in Croatia in the One-Health Context" for consideration for publication in the Pathogens.

We have carefully considered the reviewer’s comments and revised the manuscript accordingly. The changes are marked using “Track changes” in the revised version of the manuscript. Below are the answers to specific reviewers’ comments.

I checked the revised manuscript submitted by the authors. Most of the issues have been resolved carefully by the authors. I like the idea that the authors replace the table with a graph figure which gives new appearance of the results. 

I found also the authors have included and/or changed, some, not all of the suggested references. I have few minor suggestions

a) Introduction: page 1 lines 43-45 " Zoonotic potential of HEV genotypes 3 and 4 has been well demonstrated for pigs [5] and wild boars [6], as well as for the genotype 3 in rabbits [7] and deer [8]. Please also include the other potential animal reservoirs such as cows (PMID: 27286751), goat (PMID: 33322702, PMID: 32659521) and sheep (PMID: 33322702). Please include the above information and cite the suggested references.

We have inserted the proposed information and references in our review.

2- Introduction page 2 line 48-49: The cell culture system has been established for genotype 3 and 4 HEV strains, but genotype 1 replicate poorly in vitro [12]. Please add that. However, HEV genotype 1 can replicate efficiently in primary cells (  PMID: 31727684, PMID: 32316431, PMID: 32455708, PMID: 30420629, PMID: 32824088) or in vivo animal models (PMID: 26038457). Please add the information with all suggested references.

We thank the reviewer for all the suggestions. All proposed references are included.

3- Introduction page 2 lines 64-66:  In addition, extrahepatic manifestations of HEV have been increasingly recognized with neurological (55%), cardiovascular or hematological (35%), and gastrointestinal 66
manifestations (7%) being the most common [26]

  1. a) Please delete cardiovascular, instead include renal . In addition please add complication during pregnancy.

  2. b) Please delete all the percentage, they are not true.

  3. c) Please include the 2 other references that cover all extrahepatic manifestation of HEV (PMID: 27913223 )

We have replaced cardiovacular with renal extrahepatic manifestation as they appear more often. New references are added. Percentages are deleted. We thank the reviewer for pointing out the pregnancy complications in HEV infected women, we added that to introduction part of review.

Please  correct/ add the information with all suggested references.

At the end, I want to thank the authors for their patience and including all the suggested comments. Congratulations for this nice review.  

In conclusion, we thank the reviewer for the useful and constructive comments.

We hope that we have improved the consistency, clarity and interpretation of data in the revised manuscript and that it will be suitable for publication in the Pathogens.

Thank you again for the privilege of submitting our work to the Pathogens.

Sincerely,

Anna Mrzljak, MD, PhD

University Hospital Center Zagreb

School of Medicine, Zagreb, Croatia